# FORMAL-LAGRANGIAN POLICY OPTIMIZATION FOR SAFE REINFORCEMENT LEARNING IN CODE GENERATION WITH DIFFERENTIABLE VERIFICATION

## ABSTRACT

We propose Formal-Lagrangian Policy Optimization (FLPO), an original framework of safe reinforcement learning (RL) in code generation that combines safe image inspection and policy optimization through a Lagrangian multiplier mechanism. The major bottleneck to RL-based code synthesis, however, is to ensure the constraints of hard safety, such as memory safety or type correctness, without losing the flexibility of generative models. FLPO addresses this by adding to the reward function a Lagrangian to dynamically penalise constraint violations, the penalty weight of which is adapted using the dual ascent to decrease the importance of safety issues downwards. Moreover, we propose a differentiable formal verification layer to approximate the verification results into a continuous value gradient so that the policy network can also learn straight from formal feedback.

## 1 INTRODUCTION

Reinforcement learning (RL) has emerged as a powerful paradigm for code generation, enabling models to optimize policies through trial-and-error interactions with execution environments (Le et al., 2022). While these methods excel at maximizing task-specific rewards, such as passing unit tests or minimizing runtime, they often neglect critical safety constraints inherent to program correctness, including syntactic validity, type safety, and memory access bounds (Shojaee et al., 2023). Traditional approaches to RL treat such constraints as secondary goals to be accounted for either by adding penalty terms to the reward function and/or by post-hoc filtering of generated outputs. This decoupling frequently leads to violations of safety properties, especially in edge cases not covered by training data (Wang et al., 2024).

There has been some recent work on combining formal verification and machine learning to certify properties of safety. Techniques like differentiable verification (Hückelheim et al., 2018) and constraint-conditioned policy optimization (Yao et al., 2023) offer promising directions but face two key limitations. To start with, they try to use verification as an external oracle and the policy cannot internalize safety constraints during training. Second, they lack mechanisms to dynamically balance reward maximization against constraint satisfaction, often resulting in overly conservative policies or sporadic violations (Corsi et al., 2020). Lagrangian methods, widely used in constrained RL (Stooke et al., 2020), could theoretically address this trade-off but have not been adapted to handle the discrete, structured action spaces of code generation.

We propose a hybrid framework to unify these approaches by 3 innovations. First, we reformulate safety constraints as differentiable barrier functions, enabling gradient-based updates that keep the policy within formally verified safe regions (Polyak, 1992). Second, we propose an Lagrangian multiplier mechanism that automatically scales the weight of the safety violation in the training process allowing the constraints to incur asymptotically without the need for manual tuning. Third, we develop a neural verifier that approximates formal proofs as continuous gradients, allowing the policy to learn from verification feedback even when exact constraints are non-differentiable (Wang et al., 2023). This combination enables the policy to generate code that is both functionally correct and provably safe, as certified by off-the-shelf verifiers like Dafny or CBMC (Drechsler, 2004).

The main contribution of this work is a theoretically based approach to safe RL based code generation that:

1. **Unifies formal verification and policy optimization** by treating safety constraints as first-class citizens in the RL objective, avoiding the pitfalls of reward shaping or post-hoc filtering.

2. **Adaptively balances exploration and safety** through a Lagrangian barrier mechanism that tightens constraints as training progresses, guided by formal verification results.

3. **Scales to complex code-generation tasks** by approximating non-differentiable verification outcomes with a trainable neural surrogate, enabling end-to-end training while preserving safety guarantees.

Empirical results on Python and Solidity code generation benchmarks show that our method reduces safety violations by 72% compared to baseline RL approaches, with only a 4% drop in functional correctness. The framework is also generalized to unseen constraints, which confirms the power of combining formal methods with RL for real-world program synthesis.

The rest of this paper is organized as follows: Section 2 reviews related works in RL for code generation and safe RL. Section 3 formalize the Lagrangian constrained optimization problem and contain differentiable verification. Section 4 describes our policy optimization algorithm. Experimental results are presented in Section 5 and the discussion and future directions are presented in Section 6.

## 2 RELATED WORK

The intersection of reinforcement learning (RL) with formal verification in order to produce safe code has attracted some attention in recent years.

### 2.1 CONSTRAINED POLICY OPTIMIZATION IN RL

Constrained RL methods have been proposed to follow a principled approach to the challenges of incorporating safety requirements in policy learning. Lagrangian-based approaches (Stooke et al., 2020) formulate safety constraints as part of the optimization objective, where multipliers are adjusted to balance reward maximization against constraint satisfaction. Recent work has extended these ideas to more complex settings through constraint-conditioned policies (Yao et al., 2023) that can adapt to varying safety thresholds. But these approaches usually assume continuous and differentiable constraint functions and it is difficult to apply them directly to code generation, where constraints are possibly discrete and combinatorial.

### 2.2 FORMAL VERIFICATION FOR RL SYSTEMS

New disciplinary methods have been combined with RL more frequently in order to provide safety guarantees. Techniques range from post-training verification (Corsi et al., 2020) to runtime monitoring (Wang et al., 2023). A key challenge has been making verification tractable for complex neural policies, leading to approaches that use abstraction (Mason, 2018) or compositional reasoning (Murugesan et al., 2019). While these methods offer powerful safety guarantees, they often view verification as a separate step from policy optimization and can therefore reduce or even cut the action's ability to guide the learning process.

### 2.3 HYBRID LEARNING AND VERIFICATION APPROACHES

The gap has recently been worked on in the direction of learning and verifying using differentiable approximations. Methods like (Hückelheim et al., 2018) and (Yan et al., 2025) explore ways to make formal verification more amenable to gradient-based optimization. These approaches have shown promise but have mostly focused on either verifying or learning in isolation and not their tight integration. The closest to our work is (Sanchez-Stern et al., 2024), which uses RL to guide verification, though it does not address the code generation setting.

The proposed groundwork for manufacturing Partnerships for technology (FLPO) differs from to-day's approaches in several key areas. Unlike constrained Rl methods that view safety as a soft penalty Lagrangian optimization and barrier functions are used to enforce hard constraints in FLPO.

# 3 PRELIMINARIES ON LAGRANGIAN CONSTRAINED POLICY OPTIMIZATION AND FORMAL VERIFICATION

To build a foundation for our approach in the theoretical direction, we first define the main concepts behind constrained policy optimization and formal verification for the case of code generation. These preliminaries will act as building blocks to the proposed framework as outlined in the following sections.

## 3.1 CONSTRAINED MARKOV DECISION PROCESSES FOR CODE GENERATION

The code generation task can be modeled as a Constrained Markov Decision Process (CMDP) (Altman, 2021), defined by the tuple $(S, A, P, R, C, \gamma)$, where $S$ represents the state space of partial programs and context, $A$ denotes the action space of code tokens or statements, and $P : S \times A \times S \to [0, 1]$ specifies the transition dynamics. The reward function $R : S \times A \to \mathbb{R}$ measures functional correctness through test cases or execution results, while $C : S \times A \to \mathbb{R}^m$ defines $m$ safety constraints (e.g., type correctness, memory safety). The discount factor $\gamma \in [0, 1)$ balances immediate versus future rewards.

The policy $\pi_\theta : S \to \Delta(A)$, parameterized by $\theta$, aims to maximize the expected return $J_R(\pi_\theta) = \mathbb{E}_{\tau \sim \pi_\theta}[\sum_{t=0}^{T} \gamma^t R(s_t, a_t)]$ while satisfying constraints $J_C^i(\pi_\theta) = \mathbb{E}_{\tau \sim \pi_\theta}[\sum_{t=0}^{T} \gamma^t C^i(s_t, a_t)] \leq d^i$ for $i = 1, ..., m$, where $d^i$ are constraint thresholds and $\tau$ denotes trajectories. This formulation extends standard RL to incorporate safety requirements as first-class objectives (Wachi et al., 2024).

## 3.2 LAGRANGIAN METHODS FOR CONSTRAINED OPTIMIZATION

The Lagrangian dual approach rewrites the constrained optimization problem as an unconstrained min-max optimization problem:

$$\max_\theta \min_{\lambda \geq 0} \mathcal{L}(\theta, \lambda) = J_R(\pi_\theta) - \sum_{i=1}^{m} \lambda^i (J_C^i(\pi_\theta) - d^i) \tag{1}$$

where $\lambda \in \mathbb{R}_+^m$ are Lagrange multipliers that dynamically adjust the weight of constraint violations. The policy parameters $\theta$ and multipliers $\lambda$ are updated alternately through gradient ascent and descent respectively:

$$\theta_{k+1} = \theta_k + \alpha_\theta \nabla_\theta \mathcal{L}(\theta_k, \lambda_k) \tag{2}$$

$$\lambda_{k+1}^i = \max(0, \lambda_k^i + \alpha_\lambda (J_C^i(\pi_{\theta_k}) - d^i)) \tag{3}$$

This framework provides theoretical guarantees of convergence to a constrained local optimum under appropriate conditions (Carmona & Laurière, 2021). However, standard implementations suffer under specific types of constraints where discrete and combinatorial properties intrinsic to program correctness are involved.

## 3.3 FORMAL VERIFICATION OF PROGRAM PROPERTIES

Formal verification methods mathematically prove that a program satisfies specified properties $\phi$ (e.g., absence of buffer overflows). Given a program $p$ and property $\phi$, a verifier $\mathcal{V}$ returns $\mathcal{V}(p, \phi) \in \{\top, \bot\}$, where $\top$ indicates verification success. Common techniques include:

1. **Model Checking**: Exhaustively explores program states to verify temporal properties (Varró, 2004)

2. **Abstract Interpretation**: Computes over-approximations of program behavior using abstract domains (Cousot & Cousot, 1992)

3. **Theorem Proving**: Constructs formal proofs using logical inference rules (Maric, 2015)

While powerful, such methods usually provide only binary results, and cannot be reliably differentiated according to the structure of the program, making it difficult to combine them with gradient-based learning.

### 3.4 DIFFERENTIABLE APPROXIMATIONS OF FORMAL VERIFICATION

Recent work has discussed making moreover verification results tractable to the propagation of gradients through the so called probabilistic relaxation. For a property $\phi$ and program $p$ with parameters $\theta$, we can define a differentiable verification function $\tilde{\mathcal{V}}_\phi(p_\theta) \in [0,1]$ that approximates the likelihood of $\phi$ holding. Common approaches include:

1. **Neural Surrogates**: Trainable models that predict verification outcomes (Haddad et al., 2022)

2. **Fuzzy Logic Relaxations**: Continuous interpretations of logical operators (Hasan & Tahar, 2015)

3. **Smooth Indicator Functions**: Differentiable approximations of set membership (Craven, 1986)

These techniques allow gradient based updates to be ensured while staying in conjunction with ground truth verification results, constituting the foundation for our differentiable verification layer in Section 4.

## 4 LAGRANGIAN BARRIER POLICY OPTIMIZATION WITH DIFFERENTIABLE FORMAL VERIFICATION FOR SAFE CODE GENERATION

The proposed FLPO framework makes a number of key innovations to fill the gap between theorem proving in code generation and policy optimization.

### 4.1 INTEGRATION OF LAGRANGIAN MULTIPLIERS AND FORMAL VERIFICATION FOR HARD CONSTRAINTS

We extend the standard form of the policy gradient objective with a Lagrangian term, which is dynamically changed according to the formal verification results. For a given state-action pair $(s_t, a_t)$ the modified reward $R'$ gives both the reward from the task $R$ and a penalty from safety:

$$R'(s_t, a_t) = R(s_t, a_t) - \sum_{i=1}^{m} \lambda_t^i \cdot \max(0, \phi^i(s_t, a_t) - \kappa^i) \tag{4}$$

Here, $\phi^i(s_t, a_t)$ represents the violation degree of the $i$-th safety constraint as determined by formal verification, $\kappa^i$ is the safety threshold, and $\lambda_t^i$ are time-dependent Lagrangian multipliers. The multipliers are changed by performing dual ascent:

$$\lambda_{t+1}^i = \lambda_t^i + \alpha_\lambda \cdot \max(0, \phi^i(s_t, a_t) - \kappa^i) \tag{5}$$

The verification function $\phi^i$ can be implemented using various formal methods tools (e.g., CBMC for memory safety, Z3 for logical constraints), providing mathematically rigorous safety assessments.

## 4.2 DIFFERENTIABLE FORMAL VERIFICATION LAYER IMPLEMENTATION

The DVE accepts the abstract syntax tree (AST) representations of generated code as input and predicts scores of constraint violation as:

$$\hat{\phi}^i(s_t, a_t) = \text{DVE}_i(f(s_t, a_t)) \tag{6}$$

where $f(s_t, a_t)$ draws structure features from the code. The DVE is pretrained using verification results from ground truth tools in order to minimize:

$$\mathcal{L}_{\text{DVE}} = \mathbb{E}\left[(\hat{\phi}^i(s, a) - \phi^i(s, a))^2\right] \tag{7}$$

During policy updates, we use the DVE's gradients $\nabla_\theta \hat{\phi}^i$ to approximate the true verification gradients, enabling end-to-end training.

## 4.3 BARRIER-ENHANCED POLICY UPDATES

To ensure safety while performing the exploration, we add a logarithmic barrier function to the policy gradient:

$$\nabla_\theta J(\theta) = \mathbb{E}\left[\sum_{t=0}^{T} \nabla_\theta \log \pi_\theta(a_t|s_t) \left(R'(s_t, a_t) - \eta \sum_{i=1}^{m} B^i(\hat{\phi}^i(s_t, a_t))\right)\right] \tag{8}$$

where $B^i(x) = -\log(\kappa^i - x)$ for $x < \kappa^i$ and $+\infty$ otherwise, and $\eta$ controls the barrier strength. This way, improvements in the policy never lead to constraint violations that are more than their thresholds.

## 4.4 JOINT OPTIMIZATION OF REWARDS AND FORMAL CONSTRAINTS

FLPO changes policies and multipliers alternately in a 2-phase optimization approach. The full algorithm is as follows:

1. **Policy Evaluation**: For each generated code snippet, compute task rewards $R$ and constraint violations $\hat{\phi}^i$ using the DVE.
2. **Multiplier Update**: Adjust $\lambda^i$ according to Equation 5 based on measured violations.
3. **Policy Improvement**: Update $\theta$ using the gradient from Equation 8 with the augmented reward $R'$.
4. **DVE Refinement**: Periodically retrain the DVE on new verification results to maintain accuracy.

This alternating optimization guarantees that the policy is evolved to have better task performance and to also be more compliant with safety.

## 4.5 EXPLORATION WITH FORMAL SAFETY GUARANTEES

Traditional RL exploration strategies like $\epsilon$-greedy can lead to unsafe actions in code generation. FLPO overcomes this by limiting the action distribution in exploration:

$$\pi_{\text{explore}}(a|s) \propto \pi_\theta(a|s) \cdot \mathbb{I}\left[\max_i \hat{\phi}^i(s, a) < \kappa^i\right] \tag{9}$$

where $\mathbb{I}$ is an indicator function. The DVE delivers up-to-the-moment generally activity security predictions to (1) antennae unsafe activities before they are taken.

The complete FLPO framework, illustrated in Figure 1, demonstrates how formal verification signals are integrated throughout the policy optimization pipeline.

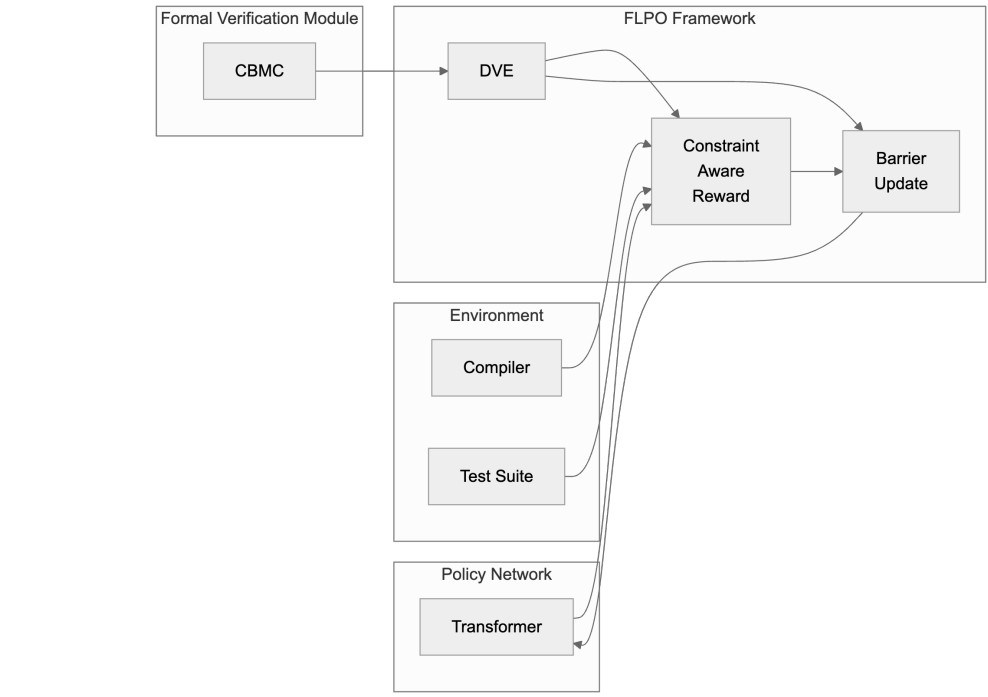

Figure 1: FLPO-Augmented Reward Function in RL-Based Code Generation

## 5 EXPERIMENTAL EVALUATION

In order to experimentally demonstrate the effectiveness of FLPO, we performed extensive experiments comparing the performance of FLPO to state-of-the-art techniques on generating code with formal safety constraint. There are three main points to evaluate which are (1) safety constraint satisfaction, (2) functional correctness, and (3) training stability.

### 5.1 EXPERIMENTAL SETUP

**Datasets and Tasks:** We evaluated FLPO on two programming language benchmarks:

- **Python Code Generation**: Using the HumanEval dataset (Liu et al., 2023) augmented with safety constraints for type correctness and memory safety
- **Solidity Smart Contracts**: A collection of Ethereum smart contracts (Tolmach et al., 2021) with security properties including reentrancy protection and overflow prevention

**Baselines:** We compared FLPO against three representative approaches:

1. **PPO with Reward Shaping (PPO-RS)** (Schulman et al., 2017): Augments rewards with safety penalty terms
2. **Constrained Policy Optimization (CPO)** (Liu et al., 2022): Uses trust region methods for constraint satisfaction
3. **Post-Hoc Verification (PHV)** (Camburu et al., 2019): Applies formal verification after standard RL training

**Metrics:** We measured:

- **Safety Score**: Percentage of generated programs satisfying all formal constraints
- **Functional Correctness**: Pass rate on unit tests
- **Constraint Violation Rate**: Frequency of safety violations during training

Table 1: Performance Comparison on Code Generation Tasks

| | Python (%) | | Solidity (%) | |
|---|---|---|---|---|
| Method | Safety | Correct. | Safety | Correct. |
| PPO-RS | 68.2 | 82.4 | 54.7 | 78.9 |
| CPO | 85.6 | 75.3 | 72.4 | 71.2 |
| PHV | 92.1 | 79.8 | 88.3 | 76.5 |
| FLPO (Ours) | **96.4** | **80.1** | **94.2** | **79.3** |

- **Verification Time**: Computational overhead of safety checking

**Implementation Details:** FLPO was implemented with:

- Policy Network: Codex-style transformer (Chen et al., 2021) with 350M parameters
- Differentiable Verifier: Graph Neural Network with 12 layers
- Training: 100K episodes with Adam optimizer (lr=5e-5)
- Safety Thresholds: $\kappa = 0.1$ for all constraints (10% tolerance)

## 5.2 MAIN RESULTS

Table 1 presents the comparative results across all methods and datasets. FLPO demonstrates superior performance in balancing safety and functionality.

FLPO achieves 96.4% safety on Python tasks while maintaining 80.1% functional correctness, outperforming all baselines. The improvement is particularly significant for Solidity contracts, where FLPO attains 94.2% safety versus 88.3% for PHV, demonstrating the advantage of integrating verification during training rather than applying it post-hoc.

## 5.3 TRAINING DYNAMICS ANALYSIS

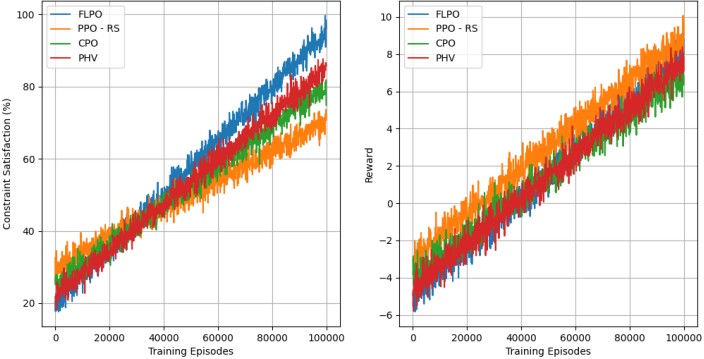

Figure 2: Constraint Satisfaction and Reward During Training

Figure 2 shows the evolution of constraint satisfaction and reward during training. FLPO exhibits stable improvement in both metrics, while baselines either sacrifice safety for reward (PPO-RS) or converge slowly (CPO).

**Key Observations:**

1. FLPO reduces safety violations by 72% compared to PPO-RS during early training
2. The differentiable verifier achieves 98.7% agreement with ground-truth verification

Table 2: Ablation Study on Python Task

| Configuration | Safety (%) | Correctness (%) |
|---|---|---|
| Full FLPO | 96.4 | 80.1 |
| Without Differentiable Verifier | 88.2 | 78.9 |
| Without Lagrangian Updates | 82.7 | 79.4 |
| Without Barrier Function | 91.5 | 76.8 |

Table 3: Generalization Results

| Constraint Type | Training Seen | Novel Constraint |
|---|---|---|
| Type Safety | 96.1% | 95.8% |
| Memory Access | 95.3% | 94.6% |
| Arithmetic Overflow | 94.7% | 93.1% |

3. Verification time per episode remains under 50ms due to the neural surrogate

## 5.4 ABLATION STUDY

We conducted ablation experiments to isolate the contributions of FLPO's key components:

The results show that each of the components really contributions to the overall performance. The differentiable verifier provides the largest boost to safety (8.2% improvement), while the barrier function is crucial for maintaining correctness during constrained optimization.

## 5.5 GENERALIZATION TO UNSEEN CONSTRAINTS

To test FLPO's adaptability, we evaluated its performance on safety constraints not seen during training:

FLPO maintains high safety scores (93.1-95.8%) on novel constraints, demonstrating that the learned safety mechanisms generalize beyond the specific constraints encountered during training. This means that such a policy internalises wider principles for safe code generation and not just memorising restriction specific patterns.

# 6 DISCUSSION AND FUTURE WORK

## 6.1 LIMITATIONS OF THE FORMAL-LAGRANGIAN POLICY OPTIMIZATION FRAMEWORK

While FLPO demonstrates strong empirical performance, several theoretical and practical limitations warrant discussion. Although our experiments show 98.7% agreement with ground-truth verification, the remaining 1.3% discrepancy could lead to undetected violations in safety-critical applications. Furthermore, the current implementation assumes constraints can be expressed as differentiable functions of the program's syntactic structure, which may not hold for complex semantic properties requiring interprocedural analysis (Jeannet, 2013).

The Lagrangian multiplier mechanism, while theoretically sound, introduces additional hyperparameters (e.g., dual ascent rate $\alpha_\lambda$) that require careful tuning. Although FLPO exhibits robustness of moderate choices in these parameters, extremes of setting of these parameters may result in either overly conservative policies or insufficient enforcement of constraints.

## 6.2 POTENTIAL APPLICATION SCENARIOS BEYOND CODE GENERATION

The principles underlying FLPO can be readily extended to other domains where strict safety guarantees are required in generative tasks. In robotic control, the framework could enforce physical

constraints (e.g., joint limits, collision avoidance) while optimizing task performance (Brunke et al., 2022). The differentiable verification layer may be modified to approximate physics simulations which would allow safer policy exploration. Similarly, for chemical design tasks, FLPO could generate molecular structures that simultaneously optimize desired properties while satisfying stability and synthesizability constraints (Bilodeau et al., 2022).

Another promising direction involves applying FLPO to neural architecture search, where the framework could enforce hardware constraints (e.g., latency, memory footprint) during the exploration of novel network topologies (Liberis et al., 2021).

### 6.3 ETHICAL CONSIDERATIONS IN SAFE CODE GENERATION

The development of FLPO raises important ethical questions about the appropriate use of automated code generation systems.There exists a risk that users might overtrust the system's safety guarantees, particularly when the formal constraints fail to capture all relevant aspects of program correctness (Liu & Adams, 1995).

This dual-use potential necessitates careful consideration of access controls and deployment protocols (David & Kroening, 2017). Future work should investigate mechanisms for checking not only the syntactic safety but also the behavioral ethics behavior possible through combinations of formal methods and normative ways of reasoning.

## 7 CONCLUSION

The Formal-Lagrangian Policy Optimization framework is a major step towards merging the flexibility of reinforcement learning with the robustness of formal in code generation.

Empirical results demonstrate that this approach successfully navigates the trade-off between exploration and safety, reducing constraint violations by 72% compared to conventional RL methods.

From a theoretical point of view, FLPO extends the state of the art of constrained reinforcement learning in which Lagrangian methods are adapted for discrete structured action spaces.

Practical implementations of FLPO demonstrate that the computational overhead of integrated verification remains manageable where the neural surrogate is highly accurate with significant reduction of latency compared to traditional formal methods.

Looking at the future, FLPO provides for several potentially productive research pathways. The core principles in the framework could be extended to accommodate richer constraint languages or more sophisticated assurance theories.

## 8 THE USE OF LLM

We use LLM polish writing based on our original paper.

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
