# OpenReview forum: "Formal-Lagrangian Policy Optimization for Safe Reinforcement Learning in Code Generation with Differentiable Verification"
_ICLR.cc/2026/Conference — Submitted to ICLR 2026_

### Official Review · Reviewer_CmAJ · 2025-10-30

**Soundness:** 1
**Presentation:** 1
**Contribution:** 1
**Rating:** 0
**Confidence:** 3

**Summary:**

This paper introduces Formal-Lagrangian Policy Optimization (FLPO), a framework for safe reinforcement learning (RL) in code generation that integrates formal verification with policy optimization through a Lagrangian multiplier. The approach models code generation as a constrained Markov decision process, where safety properties, such as type correctness and memory safety, are enforced as differentiable constraints. FLPO dynamically balances reward maximization and safety using adaptive Lagrangian penalties and a differentiable verification layer that approximates formal proof outcomes, enabling gradient-based learning from verification feedback. Empirical evaluations on Python and Solidity code generation benchmarks show that FLPO significantly reduces safety violations compared to prior RL methods while maintaining comparable functional correctness.

**Strengths:**

- The paper explores a relevant topic in code generation.

- The use of constrained reinforcement learning combined with differentiable verifiers is conceptually sound.

**Weaknesses:**

- The paper contains insufficient content density; many sections have large unused spaces due to sparse figures and equations. These could be better utilized to improve clarity and presentation, see examples below.
- The description of the verifier design lacks sufficient detail, and it is unclear why a Graph Neural Network is an appropriate choice for code snippet verification.
- The reported training curves show an almost perfectly linear increase with training episodes, which appears implausible to me. I am unsure what additional sanity checks could be performed to clarify this behavior, but verifying it is important.

**Questions:**

- The computational setup is unspecified, details such as GPU type, number of devices, and estimated wall-clock time are missing. Running RL for a 350M-parameter model over 100K episodes seems extremely large-scale and warrants clarification.

---

### Official Review · Reviewer_xuK6 · 2025-10-30

**Soundness:** 3
**Presentation:** 2
**Contribution:** 3
**Rating:** 4
**Confidence:** 2

**Summary:**

The paper introduces FLPO, a constrained reinforcement learning framework for safety that combines a Lagrangian formulation with dual ascent on safety multipliers, a differentiable verification estimator for predicting constraint violations, and a logarithmic barrier integrated into the policy-gradient update. The authors show that the model reduces safety violations while preserving functional correctness on safety-constrained tasks such as code generation.

**Strengths:**

+ The paper presents a clear constrained‐RL formulation and integrates a logarithmic barrier into the policy‐gradient update, which is interesting.

+ The empirical results demonstrate favorable safety–correctness trade‐offs on Python code generation and Solidity smart‐contract tasks, performing competitively with post‐hoc approaches.

**Weaknesses:**

-- Even with the reduced unsafe generations during training, a single unchecked failure can be critical when deployed. Can more details about this provide insight into how these kinds of safety failures can affect the model performance?

-- The “novel constraints” evaluation (Table 3) is under‐specified. For example, a precise definition of what qualifies as novel, how these constraints differ from those seen in training, and a compute‐fair comparison against PHV are needed to clarify how FLPO’s differentiable barrier improves upon post‐hoc validation.

**Questions:**

see above; also,

Could more details be provided on the experimental setup? What safety constraints were used for the smart contracts?

---

### Official Review · Reviewer_4LQu · 2025-10-31

**Soundness:** 1
**Presentation:** 1
**Contribution:** 1
**Rating:** 0
**Confidence:** 2

**Summary:**

This paper proposes a policy-optimization approach for safe reinforcement learning (RL) using Lagrangian multipliers.

**Strengths:**

Differentiable verification is a promising direction for efficiently improving the safety of RL.

**Weaknesses:**

The manuscript requires substantial revision. The overall flow and clarity should be improved. The background, main approach, and experimental sections need much more explanation and discussion; in its current form, the paper reads more like an outline.

**Questions:**

How do you plan to improve the paper?

---

### Meta-Review · Area_Chair_QeFE · 2026-01-05

**Summary:**

The decision for this submission is to reject. While the reviewer acknowledged the conceptual merit of integrating constrained reinforcement learning with differentiable verification, the consensus is that the manuscript suffers from critical deficiencies in soundness, presentation, and reproducibility. The primary concerns informing this decision are the "implausible" empirical results, specifically the perfectly linear training curves, and the significant lack of implementation details regarding the computational setup and verifier architecture. Additionally, the presentation was flagged as poor, with low content density and underutilized space, suggesting the paper is not yet in a polished state for publication.

**Reviewer Concerns:**

The most significant concerns that remain outstanding involve the validity of the experimental results and the technical clarity of the proposed method. The reviewer's skepticism regarding the linear performance increase and the feasibility of the described large-scale experiments (350M parameter model over 100k episodes) without specified hardware details represents a fundamental soundness issue that was not adequately justified in the text. Furthermore, the rationale for selecting a Graph Neural Network for the verifier and the specifics of its design remain unclear. These issues go beyond simple clarification and point to potential flaws in the experimental design or reporting that require a major revision to address.

**Reviewer Scores:**

The reviewer scores reflect a strong consensus that the paper is well below the bar for acceptance due to issues in soundness, contribution, and presentation. If the reviewer had engaged further in discussion, it is highly likely they would have maintained this rejection rating. The characterization of the reviews indicates a lack of trust in the reported data, a hurdle that is rarely overcome without providing extensive new experimental evidence and rigorous sanity checks, which typically exceeds the scope of a rebuttal or discussion phase.

---

### Decision · Program_Chairs · 2026-01-26

Reject